# Are Reactive Oxygen Species (ROS) the Main Mechanism by Which Copper Ion Treatment Degrades the DNA of *Mycobacterium avium* subsp. *paratuberculosis* Suspended in Milk?

**DOI:** 10.3390/microorganisms10112272

**Published:** 2022-11-16

**Authors:** Marcela Villegas, Carlos Tejeda, Reydoret Umaña, Esperanza C. Iranzo, Miguel Salgado

**Affiliations:** 1Instituto de Medicina Preventiva Veterinaria, Facultad de Ciencias Veterinarias, Universidad Austral de Chile, Saelzer Building 5° Floor, Campus Isla Teja, Valdivia P.O. Box 567, Chile; 2Facultad de Ciencias, Escuela de Graduados, Universidad Austral de Chile, Valdivia P.O. Box 567, Chile; 3Facultad de Ciencias Veterinarias, Escuela de Graduados, Universidad Austral de Chile, Valdivia P.O. Box 567, Chile; 4Instituto de Ciencia Animal, Facultad de Ciencias Veterinarias, Universidad Austral de Chile, Valdivia P.O. Box 567, Chile

**Keywords:** *Mycobacterium avium* subsp. *paratuberculosis*, copper, ROS, copper chelators, ROS quenchers, DNA protection, milk

## Abstract

Background: *Mycobacterium avium* subsp. *paratuberculosis* (MAP) is the causal agent of paratuberculosis. This pathogen is able to survive adverse environmental conditions, including the pasteurization process. Copper, a well-studied metal, is considered an important antibacterial tool, since it has been shown to inactivate even MAP in treated milk through unknown mechanisms. The aim of the present study is to show the effect of copper ions, and reactive oxygen species (ROS) generated in response to oxidative stress, on the damage to MAP DNA when exposed to a copper ion challenge in cow’s milk. Methodology: Spiked milk with different MAP bacterial loads was supplemented with blocking agents. These were either the copper chelators ethylenediaminetetraacetic acid (EDTA) and batocuproin (BCS) or the ROS quenchers D-mannitol, gallic acid and quercetin. The DNA protection, MAP viability and ROS production generated after exposure to a copper challenge were then measured. Results: In a bacterial load of 10^4^ cells mL^−1^, blocking effects by both the copper chelators and all the ROS quenchers offered significant protection to MAP DNA. In a concentration of 10^2^ cells mL^−1^, only D-mannitol and a mix of quenchers significantly protected the viability of the bacteria, and only at a concentration of 10^6^ cells mL^−1^ was there a lower production of ROS when supplementing milk with gallic acid, quercetin and the mix of quenchers. Conclusion: Based on these findings, it may be concluded that MAP DNA damage can be attributed to the combined effect of the direct copper ions and ROS generated. Nevertheless, taking into account the antioxidant environment that milk provides, the direct effect of copper could play a prominent role.

## 1. Introduction

*Mycobacterium avium* subsp. *paratuberculosis* (MAP) is the causal agent of paratuberculosis, a chronic and incurable infectious disease that is widespread among domestic ruminants [1]. This infectious disease has a great economic impact [2] and MAP poses a potential threat to public health, due to its potential zoonotic link [3]. The control of this infection in herds has not been fully achieved [4] and the spread of the pathogen through milk has been a neglected issue that could, in part, explain the failure of control programs [5]. The pasteurization of milk, as a decontamination tool at cattle herd level, has not been widely used, due to its high cost and, in the case of MAP, uncertainty of its overall effectiveness [6,7]. Alternatively, the use of copper as a decontamination tool seems promising, as it exhibits an intrinsic antimicrobial effect [8]. The antibacterial efficacy of copper is very well documented in different bacterial models [8,9], including members of the *Mycobacterium* genus such as *Mycobacterium tuberculosis* [10], *Mycobacterium avium* [11] and recently in *Mycobacterium avium* subsp. *paratuberculosis* (MAP) [5,12,13]. Researchers [5,13] have provided in vitro evidence for the copper-ion-induced inactivation of MAP cells in raw cow’s milk. However, the mechanisms of copper toxicity that would explain the bacterial inactivation are not entirely clear. A widely accepted mechanism is the oxidative damage of important cellular components, such as lipids, proteins, cell membranes and DNA, through ions (Cu^+1^/Cu^+2^) [14,15] and/or the generation of reactive oxygen species (ROS) [8], which are highly reactive molecules, formed by the incomplete degradation of oxygen. Biologically relevant ROS are superoxide anion (O_2_^−^), hydrogen peroxide (H_2_O_2_) and the hydroxyl radical (-OH) [16]. Within a cell, the production and elimination of ROS is well balanced, but under stressful circumstances, such as copper-induced stress, this balance can be disturbed, triggering a state of oxidative stress that alters many cellular functions and structures [17]. In the particular case of MAP, evidence has recently been shown that ionic copper in a standard liquid matrix affects DNA and cellular proteins, while generating a significant increase in ROS production [18]. This would suggest that, in the case of this particular pathogen, we could point to ROS as a relevant agency for the antibacterial effects generated by copper. However, exactly how copper exerts its antibacterial effect in a complex biological matrix such as milk is still unknown. Although some authors refer to the generation of ROS as the main bactericidal mechanism [8], it is not clear how significant this is in milk, which raises the question of how important the capacity of direct redox copper ions could be in this natural liquid matrix. Therefore, the aim of the present study is to show the effect of copper ions, and ROS generated in response to oxidative stress, on the damage to MAP DNA when exposed to a copper ion challenge in cow’s milk.

## 2. Materials and Methods

### 2.1. Study Design

In order to establish the capacity of direct ionic copper and the effects of ROS generated in response to oxidative stress by bacteria after a copper ion challenge on the integrity of MAP DNA in milk, an in vitro experiment was carried out under controlled conditions. The study design set out to block the direct pathway of the ionic copper or the action of ROS generated by copper ions by adding a blocking reagent. The experimental unit was set at 500 mL of ultra-high-temperature commercial cow’s milk (UHT) containing a known concentration of MAP cells and the treatment consisted of the supplementation of a blocking reagent to each unit—either copper chelators or ROS quenchers—which was then challenged with copper ions for 30 min. The experiment took in a total of 21 experimental units. Each experimental unit was separately inoculated with a different concentration of MAP cells to mimic a low, medium or high concentration in a natural infection as follows: (A) 10^2^ cells mL^−1^; (B) 10^4^ cells mL^−1^ and (C) 10^6^ cells mL^−1^. The treatments were assigned with a number as follows: (I) ethylenediaminetetraacetic acid (EDTA); (II) batocuproin (BCS); (III) EDTA plus BCS; (IV) D-mannitol; (V) gallic acid; (VI) quercetin and (VII) D-mannitol plus gallic acid and quercetin. In addition, three positive control units were included, which consisted of milk containing each separate MAP cell concentration (10^2^, 10^4^ and 10^6^ cells mL^−1^), and which were challenged with copper ions but without a blocking reagent. Additionally, in order to rule out a direct effect of the supplemented reagents on MAP, negative controls were carried out for each chelator/quencher agent, which were incubated with MAP cells for 30 min at room temperature without a copper ion challenge. Each experimental unit was run in 3 replicates. The experimental work was carried out at the Laboratorio de Enfermedades Infecciosas, Instituto de Medicina Preventiva Veterinaria (UACh).

### 2.2. The MAP Strains and Inoculum Preparation

The MAP strain ATCC 19698 was cultured in enriched broth 7H9, as previously described [19]. In short, MAP growth was monitored weekly, using a Helios Gamma1 spectrophotometer (Thermo Scientific, Waltham, MA, USA). When the absorbance at 600 nm reached a value of 1.0, it was estimated to be in late exponential growth at a concentration of ~10^8^ MAP cells mL^−1^ with minimal dead cells present [20]. The MAP cell numbers were more precisely estimated taking into account the genome equivalent principle [21] according to a published protocol [12]. The MAP cultures were declumped by vortexing with sterile 3 mm glass beads. Then, ten-fold serial dilutions of MAP were made in sterile water and three dilutions (10^6^, 10^4^ and 10^2^ cells mL^−1^) were used to spike the milk.

### 2.3. Reagents

Milk supplementation with copper chelators or ROS quenchers was carried out using powder reagents. The concentrations used were: 20 mM EDTA (Titriplex^®^ III, Merck, Rahway, NJ, USA); 20 μM BCS (Sigma-Aldrich, St. Louis, MO, USA); 20 mM D-mannitol (Calbiochem^®^); 0.1 mg mL^−1^ gallic acid (Sigma-Aldrich) and 10 µM quercetin (Sigma-Aldrich). Quercetin was dissolved in 5 mL of absolute ethanol before being added to the milk. All other reagents were directly added.

### 2.4. Copper Ion Challenge

The MAP exposure to copper ions was performed using the same treatment protocol reported by researchers [5,13], which used a device containing two high-purity copper plates (99%) that were stimulated with a low-voltage electrical current (24 V and 3 Amperes) for 30 min, in order to stimulate a greater release of copper ions than without an electrical current. The device was inserted into a glass container (Pyrex^®^ beaker) that contained 500 mL of UHT retail milk, spiked with a known MAP concentration, under constant agitation. Aliquots were obtained from the liquid matrix before and after the copper challenge for subsequent tests.

### 2.5. Selective Separation of MAP in Milk Samples

In order to select, concentrate and separate MAP from other non-target bacteria and inhibitors possibly present in milk, subsamples of this matrix were subjected to peptide-mediated magnetic separation (PMS) from aliquots obtained before and after the copper ion challenge. Briefly, this technique uses paramagnetic beads (Dynabeads^®^ MyOne™ Tosylactivated, Thermo Fisher) coated with two biotinylated peptides (aMp3 and aMptD) with specific binding to MAP, maximizing its capture and avoiding the non-specific recovery of other mycobacteria. Peptide-mediated magnetic separation was performed on 1 mL aliquots of milk in a BeadRetriever™ (Invitrogen, Waltham, MA, USA) according to the protocol published [22].

### 2.6. The MAP Total Quantification and Viability Assessment

In order to estimate the MAP bacterial load in milk before and after the copper ion challenge, MAP DNA was extracted according to a published protocol [23]. This extraction method is based on chemical disruption using lysis buffer (2 mM EDTA, 400 mM NaCl, 10 mM Tris-HCL, pH 8.0, 0.6% SDS) and 2 μL proteinase K (10 μg μL^−1^) and mechanical disruption by a cell disrupter MiniBeadbeater-8™ (BioSpec Products Inc., Bartlesville, OK, USA). Then, the templates were quantified using a NanoDrop 2000 spectrophotometer (Thermo Scientific). Subsequently, the obtained template was subjected to a qPCR protocol, based on the IS*900* sequence detection, in a QuantStudio™ 3 system (ThermoFisher) [23]. The probe and primer sequences used were the same as those reported by researchers [12]. The number of DNA copies derived from the qPCR was calculated using a standard curve and expressed as bacterial cell equivalents (BCE) according to the previously published protocol [21]. To assess the viability of MAP in milk for each treatment after the copper challenge, a magnetic phage separation (PhMS) was performed, according to a novel published protocol [24]. This technique captures and concentrates MAP cells using 15 μL of tosylactivated paramagnetic beads attached to mycobacteriophage D29, which adheres to the cell wall of the mycobacteria and allows its capture and separation by BeadRetriever™ (Invitrogen). This technique exploits the ability of the virus to naturally infect, replicate, and then lyse only viable mycobacterial cells, thus providing DNA for molecular confirmation by IS*900* qPCR.

### 2.7. Evaluation of the Production of Reactive Oxygen Intermediates (ROS)

The production of ROS generated by MAP cells was evaluated in each of the different treatments. The non-polar fluorescent 2′,7′-dichlorodihydrofluorescein diacetate (DCFH-DA) probe (Sigma-Aldrich) was used according to a published protocol [25] with modifications [18]. DCFH-DA was converted by cellular esterase to dichlorodihydrofluorescein (DCFH) and then potentially oxidized by intracellular ROS and other peroxides, generating dichlorofluorescein (DCF). The fluorescent DCF production was measured by a spectrofluorometer (Varioskan^®^Flash Thermo Fisher) with excitation wavelengths of 485 nm and emission of 530 nm. The ROS concentration is expressed by fluorescence intensity. Each determination was run in 3 replicates.

### 2.8. Evaluation of Physicochemical Milk Properties

The physicochemical variations in the milk, produced by the effect of the treatments and copper ion challenge, were studied through the measurement of pH (pHmeter Orion, model 420A, Thermo Scientific, Waltham, MA, USA), electrical conductivity (EC) (Hanna Instrumental, edge ™ Woonsocket, RI, USA) and concentration of dissolved oxygen ([O_2_]) (Oxy 730 InoLab). Each determination was run in 3 replicates.

### 2.9. Assessing Copper Concentration in Milk

To determine the total copper concentration, the milk was calcined in a laboratory muffle (Barnstead Thermolyne™, Ramsey, MN, USA) to reach 450 °C for a total of ten hours and then digested with concentrated HCL (37% *w*/*w*) and HNO_3_ (eq L^−1^) according to a modified protocol [26]. The reading was taken by an atomic absorption spectrophotometer (AAS) and the result was reported in mg L^−1^.

### 2.10. Statistical Analysis

The reported response variable was the percentage of DNA protection (% DNA_P_) calculated from the final bacterial load (post copper challenge), taking into account the initial bacterial load (prior to copper challenge) for each experimental unit. The result of each treatment (from I to VII) was compared with the corresponding positive control in order to determine its protective effect on MAP. To determine viability, the response variable was reported as the percentage of protection of viable cells (% viability). The calculation and comparison were similar to those detailed for DNA protection. To determine significant differences, a one-way ANOVA was used in the response variable % DNA_P,_ and a Mann–Whitney U test in the response variable % viability. For the analysis of ROS production, the logarithmically transformed variable was used. Comparison of ROS production before and after the copper ion challenge was performed with a paired t-test and the analysis of ROS production in each treatment compared to the positive control after the copper ion challenge was performed using t-Student. For the copper concentration and the physicochemical properties of the milk in each treatment before and after the challenge with copper, a descriptive analysis was performed. The analyses were performed using the R program version 3.1.2 (R Development Core Team 2015). A *p*-value of <0.05 was considered significant.

## 3. Results

### 3.1. Estimation of Bacterial Load of Protected MAP Cells

None of the reagents nor ethanol had any effect on DNA integrity or MAP viability by themselves (data not shown). When milk was inoculated with 10^2^ MAP cells mL^−1^ and exposed to the copper challenge, gallic acid, quercetin and the ROS quenching mix composed of D-mannitol + gallic acid + quercetin demonstrated a significant protection of MAP DNA, compared to the control (*p* < 0.05) (Figure 1A). By contrast, all reagents, both chelators and quenchers, offered significantly greater protection compared to the control (*p* < 0.05) in the 10^4^ MAP cells mL^−1^ milk concentration (Figure 1B). Finally, in the 10^6^ MAP cells mL^−1^ milk concentration, the chelating agent EDTA, and EDTA + BCS, and the quenching agents D-mannitol, gallic acid and quercetin provided significant DNA protection compared to the control (*p* < 0.05) (Figure 1C).

When comparing the treatments with each other, significant DNA protection (*p* < 0.05) was only observed in the bacterial concentration of 10^4^ cells mL^−1^ when the ROS quenching mix was used, in contrast to D-mannitol alone. In all other cases, no significant differences were detected between treatments (*p* > 0.05).

### 3.2. Estimation of Viable Bacterial Load

Regarding MAP viability after exposure to copper ions, in the 10^2^ cells mL^−1^ bacterial load concentration, a greater protection than in the control was observed when D-mannitol and the mix of three quenchers were used (*p* < 0.05) (Figure 2A). There were no significant differences with the control (*p* > 0.05) in all other experimental units. Interestingly, viability in controls corresponding to the 10^4^ and 10^6^ cells mL^−1^ loads was not completely eliminated (Figure 2B,C). The viability was completely eliminated after exposure to copper ions only in the control corresponding to the 10^2^ cells mL^−1^ load.

### 3.3. Reactive Oxygen Species Production

For all three bacterial concentrations used, a significant increase in ROS production was observed after exposure to a copper challenge compared to no challenge for all treatments, including the positive control (*p* < 0.01). In the 10^2^ and 10^4^ cells mL^−1^ concentrations, the ROS levels were significantly higher than those in the control (*p* < 0.05) for all treatments, except for quercetin in a concentration of 10^2^ cells mL^−1^ and quercetin and a mix of quenchers in a concentration of 10^4^ cells mL^−1^ (*p* > 0.05) (Figure 3A,B). In the 10^6^ cells mL^−1^ concentration, there was greater ROS generation than in the control when BCS and EDTA+BCS supplemented the milk, and lower ROS generation when gallic acid, quercetin and a quencher mix were added (*p* < 0.05) (Figure 3C).

### 3.4. Physicochemical Changes in Milk Treated with Chelators/ROS Quenchers and Challenged with Copper

The pH of commercial UHT milk has been standardized by the manufacturer at pH 6.6. Most of the added reagents did not modify the milk’s pH, with the exception of the EDTA reagent alone and EDTA + BCS, which caused a slight acidification when added to the MAP-inoculated milk prior to the copper challenge (Table 1). Likewise, EDTA and EDTA + BCS were the only reagents that slightly increased the electrical conductivity of milk before exposure to copper (Table 1). After the copper ion challenge, an increase in pH and a decrease in both electrical conductivity and the concentration of dissolved oxygen in the spiked MAP milk was observed in all the treatments as well as in the positive control (Table 1).

### 3.5. Determination of Copper Concentration in Milk

The concentration of copper in the milk treated with chelators or quenchers prior to the copper challenge was on average 0.721 mg mL^−1^ (±0.41 SD). After the copper challenge, the concentration of this metal in the milk supplemented with chelators EDTA and EDTA+BCS was higher than it was in the control (Table 2). The copper concentration in the positive control was higher than in the experimental units that were supplemented with ROS quenchers and BCS (Table 2).

## 4. Discussion

Although we previously had some information about the antibacterial mechanism, some things still remained unclear. The present study shows evidence that explains the role of the direct effect of the copper ions, as well as the ROS generated in response to oxidative stress, on damaged DNA from MAP cells suspended in milk. It does this through the use of ionic copper blocking agents and ROS blocking agents to determine whether the interference caused by these blocking agents protected MAP DNA in milk.

The copper chelators used (EDTA and BCS) have previously shown selective chelation of Cu^+2^/Cu^+1^ [27,28], which was protective for the DNA of Gram (+) bacteria. The use of mannitol, a sugar alcohol, was justified due to its powerful quenching effects on -OH [29]. Quercetin is a flavonoid with an important antioxidant capacity against O_2_^−^ [30] and gallic acid exhibits a protective effect against oxidative damage induced by H_2_O_2_, even though it does not seem to be selective [31,32]. On the other hand, quercetin and gallic acid were selected for their distinctive antioxidant capacity against prooxidative copper [33,34].

When MAP was present in a concentration of 10^2^ cells mL^−1^ in milk, the blocking of all ROS except -OH by D-mannitol offered the highest protection against DNA compared to the positive control. In the same context, the lack of significant differences between the control and treatments I–IV may be explained by the fact that working with bacterial loads close to the detection limit of the qPCR technique for extractions from milk and by-products makes it subject to errors [35].

On the other hand, when MAP is present in milk at a concentration of 10^4^ cells mL^−1^, all blocking agents offered significant DNA protection, compared with the (+) control and there were no significant differences between them. This fact can be considered evidence that the DNA was damaged by the direct action of both copper ions and the ROS generated. Warnes and Keevil [27,28] obtained similar results using copper surfaces on other bacterial models.

Also, when the milk contained 10^6^ cells mL^−1^, the lack of significant DNA protection in treatments II and VII may be due to the fact that the BCS chelator and mix of quenchers were not present in sufficient concentrations to provide significant DNA protection against the higher bacterial load.

The production of ROS in the milk increased significantly after the copper challenge, indicating that the constant application of electrically stimulated copper favored ROS production. This is consistent with the fact that ROS are generated endogenously in aerobic bacteria as part of the microbial metabolism, and that eukaryotic cells are also present in milk [36], the production of which can drastically increase in the face of external stressors such as copper and overcome cellular antioxidant systems, causing oxidative stress [37]. The most biologically important ROS are O_2_^−^, H_2_O_2_ and -OH [8]. Their importance is due to the fact that they are known to act as direct antimicrobials and are even capable of preventing and breaking down the formation of biofilms, and some direct applications of ROS have been explored in recent years [38]. Thus, the excess of ROS in the early stages of cell death seems to be involved in the bactericidal activity of copper [28], for example, through the oxidation of cellular macromolecules such as DNA promotes cell death through apoptotic pathways [39]. Apparently, ROS as –OH are capable of directly damaging DNA through hydroxylation of free or bound deoxynucleotides (e.g., dGTP) and the sugar-phosphate bond. Due to their highly reactive nature, they also induce single- or double-strand breaks and would stimulate the formation of crosslinks that modify the double-helix structure. On the other hand, they have the potential to induce mutations directly or through protein oxidation, interfering with DNA repair or inducing error-prone repairs [17,39]. For these reasons, an effective blockade of ROS generated by exposure to copper is protective for DNA.

This idea is consistent with previous findings by researchers [13] where MAP DNA was not detected in most milk samples analyzed after exposure to copper ion treatment, suggesting extensive damage to this cellular target.

Interestingly, O_2_^−^ quenching by quercetin produced a significant decrease, or prevented a significant increase, in ROS production compared to the (+) control in all bacteria concentrations studied, perhaps due to the fact that O_2_^−^ is a primary ROS, from which H_2_O_2_ and -OH are subsequently generated dependently or independently of Fenton-type reactions (16,8]. The determination of ROS production using the nonspecific DCFH-DA probe was able to detect total fluorescence generated by the aforementioned ROS and potentially to a lesser extent by other ROS and NOS [40], but the selective quenching of ROS by means of the quenchers used conferred specificity by virtually subtracting the fluorescence contributed by each of them. In short, the significant increase in ROS production witnessed in most of the treatments compared to the respective control (+) could be directly related to the survival of MAP detected by magnetic phage separation/qPCR in all treatments at bacterial concentrations of 10^4^ cells mL^−1^ and 10^6^ cells mL^−1^, and in most at 10^2^ cells mL^−1^. This is supported by the fact that oxidative stress response mechanisms are crucial for survival [41]. If the bacterial cell continues to be viable and metabolically active, then it is capable of regulating the gene expression of antioxidant systems to prevent the damage caused by oxidative stress for some period of time, so that the mechanism of ROS production and antioxidation is maintained [42]. In MAP, these strategies would include adaptations such as enzymes of the pentose phosphate pathway, as well as KatG, SodA and GroEL, which deactivate free radicals, even protecting DNA against DNase or -OH [43,44]. The survival of MAP seen at 10^4^ cells mL^−1^ and 10^6^ cells mL^−1^, and most treatments at 10^2^ cells mL^−1^ concentrations could also be explained by its ability to mount homeostatic responses to control copper levels, like other bacteria [45].

In addition, bovine milk has been widely studied due to its antioxidant properties, which are mainly due to non-enzymatic components such as sulfur amino acids, phosphate, vitamin A and E, carotenoids, minerals such as zinc and selenium and enzyme systems such as superoxide dismutase (SOD) and glutathione peroxidase (GSHPx) [46]. For these reasons, milk, as a matrix, probably hinders the action of ROS on cellular targets. Evidence of the above is the degree of protection that milk has been shown to provide some lactobacilli against the effect of ROS when H_2_O_2_ is added [47].

In this scenario, copper could be more relevant than ROS, even surpassing components in milk, such as casein subunits, which have shown effects that favor the autoxidation of metals such as iron (Fe^+2^ and Fe^+3^), resulting in an inhibitory effect on the toxicity of this metal on certain cellular targets [48]. Furthermore, copper is capable of binding, disrupting and damaging the DNA molecule. The DNA has, on average, one copper-binding site per nucleotide, mainly at the intercalation sites on each pair of nitrogenous bases (preferably G-C sites). The high redox potential of this metal rapidly induces oxidation and ROS production in the vicinity of the binding site, favoring ROS action [14,16]. Therefore, the findings of this study indicate that by blocking the binding and action capacity of copper on biological molecules, these are protected from its direct or indirect action through the subsequent production of ROS. Finally, the concentration of copper in the milk after 30 min of exposure was similar to the concentration in phosphate-buffered saline (PBS), as previously described [19] using the same copper treatment, demonstrating a constant and efficient release of copper in milk, even though the physicochemical characteristics of cow’s milk differ from PBS. Besides, a recent published study [49] aimed to confirm or rule out whether a copper accumulation or intoxication effect was produced in calves fed with milk that was decontaminated using the same copper treatment protocol. The authors did not evidence copper toxicity based on the plasma activity of the liver enzymes evaluated, and the hepatic copper concentrations were also normal in these animals.

The present study provides information for a better understanding of the main pathways of action of copper against MAP when acting in a biological matrix, such as cow’s milk. Keeping this in mind, it is possible to adjust and design improvements in the milk decontamination device used here to guarantee its efficiency and effectively contribute to the control of MAP infection in dairy herds.

## 5. Conclusions

From the results reported in this investigation, it seems that copper may exert its antibacterial effect against MAP suspended in milk through DNA damage brought about by the complementary actions of ionic copper and the ROS produced in response to the oxidative stress generated by exposure to this metal. Considering the protective and antioxidant environment of milk, together with the results obtained through the use of ROS chelators and antioxidants, it seems that ionic copper and its oxide reduction potential have a greater relative importance in this context.

## Figures and Tables

**Figure 1 microorganisms-10-02272-f001:**
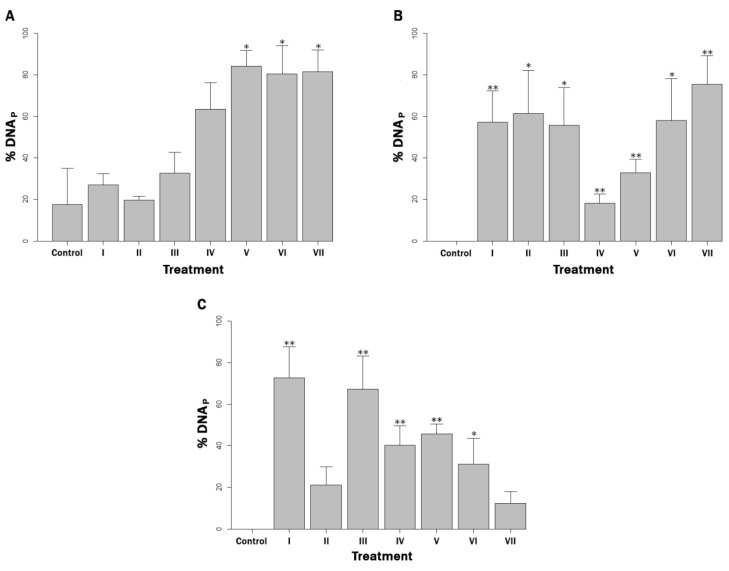
Percentage of DNA protection (% DNA_P_) in milk artificially contaminated with: (**A**) 10^2^ MAP cells mL^−1^, (**B**) 10^4^ MAP cells mL^−1^ and (**C**) 10^6^ MAP cells mL^−1^, and supplemented with (I) EDTA, (II) BCS, (III) EDTA + BCS, (IV) D-mannitol, (V) gallic acid, (VI) quercetin and (VII) D-mannitol + gallic acid + quercetin, after being challenged with copper ions for 30 min. The determination was made by PMS-qPCR. *p* < 0.05 and *p* < 0.01 are represented by * and **, respectively.

**Figure 2 microorganisms-10-02272-f002:**
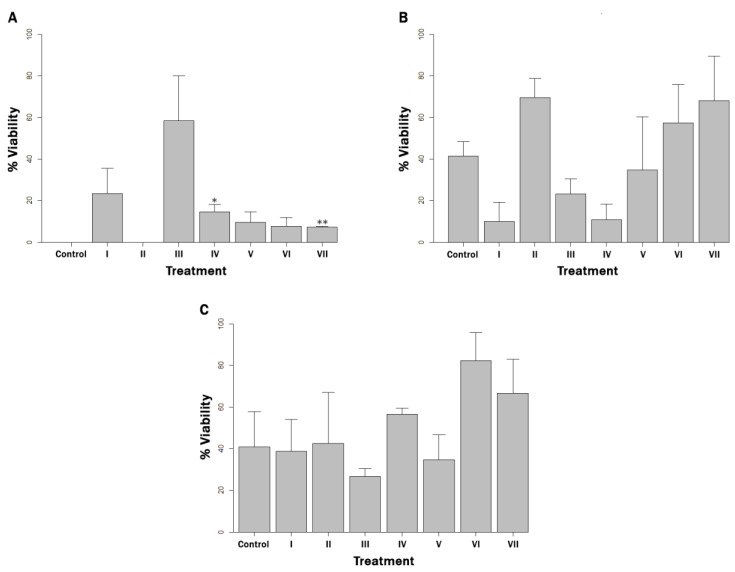
Percentage of protection of viable cells (% Viability) in milk artificially contaminated with: (**A**) 10^2^ MAP cells mL^−1^, (**B**) 10^4^ MAP cells mL^−1^ and (**C**) 10^6^ MAP cells mL^−1^, and supplemented with (I) EDTA, (II) BCS, (III) EDTA + BCS, (IV) D-mannitol, (V) gallic acid, (VI) quercetin and (VII) D-mannitol + gallic acid + quercetin, after being challenged with copper ions for 30 min. The determination was made by PhMS-qPCR. *p* < 0.05 and *p* < 0.01 are represented by * and **, respectively.

**Figure 3 microorganisms-10-02272-f003:**
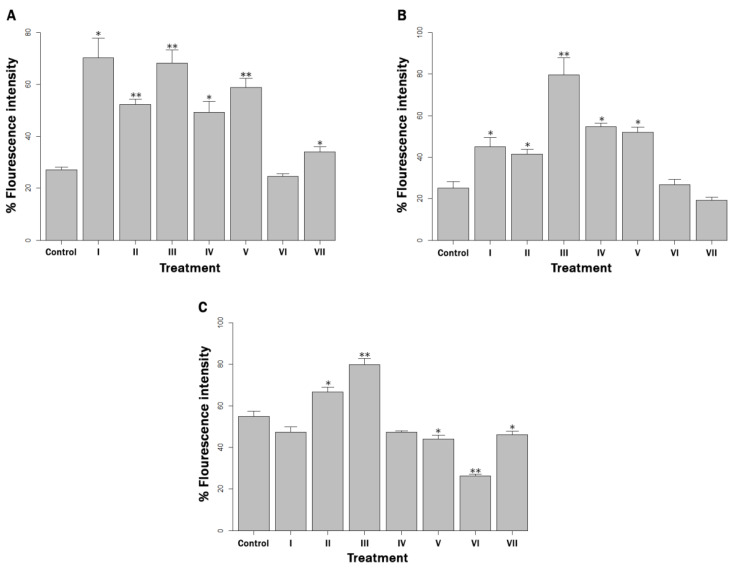
ROS production (Fluorescence Intensity) in milk artificially contaminated with: (**A**) 10^2^ MAP cells mL^−1^, (**B**) 10^4^ MAP cells mL^−1^ and (**C**) 10^6^ MAP cells mL^−1^, and supplemented with (I) EDTA, (II) BCS, (III) EDTA + BCS, (IV) D-mannitol, (V) gallic acid, (VI) quercetin and (VII) D-mannitol + gallic acid + quercetin, after being challenged with copper ions for 30 min. The determination was made using the DCFH-DA method. *p* < 0.05 and *p* < 0.01 are represented by * and **, respectively.

**Table 1 microorganisms-10-02272-t001:** Mean values for pH, electrical conductivity (EC) and the concentration of dissolved oxygen ([O_2_]) for each treatment in milk artificially contaminated with MAP before (No Cu) and after (with Cu) being challenged with copper ions for 30 min.

Treatment	pH		EC		[O_2_]	
	No Cu	with Cu	No Cu	with Cu	No Cu	with Cu
	(Mean ± SD)	(Mean ± SD)	(Mean ± SD)	(Mean ± SD)	(Mean ± SD)	(Mean ± SD)
EDTA	5.3 ± 0.0	9.2 ± 0.1	7.1 ± 0.0	6.3 ± 0.1	8.5 ± 0.1	3.1 ± 0.2
BCS	6.6 ± 0.0	9.7 ± 0.0	4.9 ± 0.1	4.0 ± 0.0	8.2 ± 0.0	2.6 ± 0.0
EDTA + BCS	5.3 ± 0.0	9.6 ± 0.1	6.5 ± 0.1	5.9 ± 0.1	8.0 ± 0.1	1.7 ± 0.3
D-mannitol	6.7 ± 0.0	9.5 ± 0.0	5.0 ± 0.0	4.1 ± 0.0	8.0 ± 0.1	3.8 ± 0.1
gallic acid	6.6 ± 0.0	9.3 ± 0.1	5.0 ± 0.1	4.1 ± 0.1	7.9 ± 0.0	4.6 ± 0.0
quercetin	6.8 ± 0.1	9.6 ± 0.0	4.9 ± 0.0	4.1 ± 0.0	9.0 ± 0.0	1.9 ± 0.2
D-mannitol + gallic acid + quercetin	6.7 ± 0.0	9.2 ± 0.0	4.9 ± 0.1	4.0 ± 0.0	7.7 ± 0.0	2.6 ± 0.0
control (+)	6.6 ± 0.1	9.7 ± 0.0	5.3 ± 0.3	4.2 ± 0.1	9.3 ± 0.2	5.5 ± 0.1

EC: Electrical conductivity expressed in mS cm^−1^; [O_2_]: Concentration of dissolved oxygen expressed in mg L^−1^; No Cu: No challenge with copper ions in the MAP-contaminated milk; with Cu: Copper plates immersed in the MAP-contaminated milk and stimulated with a low voltage (24 V) electrical current (3 Amperes) for 30 min.

**Table 2 microorganisms-10-02272-t002:** Copper concentration (mg mL^−1^) determined by AAS in milk artificially contaminated with MAP before (No Cu) and after (With Cu) the challenge with copper ions for 30 min for each treatment.

Treatment	No Cu	with Cu
EDTA	1.11	846.56
BCS	0.48	544.06
EDTA + BCS	0.77	1255.82
D-mannitol	0.50	278.28
gallic acid	0.19	378.47
quercetin	0.82	205.23
D-mannitol + gallic acid + quercetin	0.42	156.90
control (+)	1.47	624.98

No Cu: No challenge with copper ions in the MAP-contaminated milk; with Cu: Copper plates immersed in the MAP-contaminated milk and stimulated with a low voltage (24 V) electrical current (3 Amperes) for 30 min.

## Data Availability

The datasets used and/or analyzed during the current study are available from the corresponding author on reasonable request.

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
