# Peer review of "Are Reactive Oxygen Species (ROS) the Main Mechanism by Which Copper Ion Treatment Degrades the DNA of Mycobacterium avium subsp. paratuberculosis Suspended in Milk?"

_microorganisms, 2022, doi:10.3390/microorganisms10112272_

Round 1

Reviewer 1 Report (Previous Reviewer 3)

Microorganisms (Manuscript ID: microorganisms-2045097.revised), Comments to the Authors:

Title: Are reactive oxygen species (ROS) the main mechanism by which copper ion treatment degrades the DNA of Mycobacterium avium subsp. Paratuberculosis suspended in milk?

Comments

After reading the revised manuscript, I believe the authors responded to my comments and I think the manuscript can be accepted for publication. 

Reviewer 2 Report (Previous Reviewer 2)

In this new version of the manuscript, the material and methods section, was improved, better describing some experiments, as previously suggested . a Further improvement also derives from the discussion, which has been expanded to better describe the possible role of Cu and ROS, in DNA damage, and the potential toxic effects of milk treatment with Cu.

Thus, in my opinion the paper is now suitable for pubblication.

Reviewer 3 Report (Previous Reviewer 1)

The authors revised the manuscript according to the suggestions. So I think that this manuscript can be accepted in the present form.

This manuscript is a resubmission of an earlier submission. The following is a list of the peer review reports and author responses from that submission.

Round 1

Reviewer 1 Report

This article demonstrated the main mechanism by which copper ion treatment degrades the DNA of Mycobacterium avium subsp. paratuberculosis suspended in milk, which was considered to contribute to the success of eliminating the most dangerous bacteria. The experimental design and the results seem to be exquisite and valid, however, the following points would be considered to improve this article.

1.      Abbreviations. When used for the first time you write complete and abbreviation between brackets.

2.      In tables 1 and 2 the separation between the correct numbers and the fractures with a dot, not a comma.

3.      The authors did not discuss the possible side effects of copper ions residue in milk.

Reviewer 2 Report

Within this research the authors aim to investigate the effects of copper and of ROS induced by this ion against Mycobacterium avium (MAP) in cow milk. Indeed, in vitro copper ions induced inactivation of MAP in row milk has been proved, suggesting the possible use of copper instead of pasteurization that cannot guaranties complete decontamination.

To do this, milk have been inoculated with three different MAP concentration, then exposed to copper either in the presence of chelators, or ROS quencher. Subsequently, MAP DNA damage and viability have been investigated, as well as ROS production and milk properties upon Cu treatment.

Overall this straightforward work has been well conducted, the methodology is appropriate, and the results are convincing and well presented.

Some parts of Material and Methods may be described in more details, particularly the “Selective separation of MAP in milk samples” and “MAP total quantification and viability assessment” sections.

There are some typos along the text that have to be checked, and the name of microorganism in the reference list that must be written in italic font.

Reviewer 3 Report

Microorganisms (Manuscript ID: microorganisms-2008507), Comments to the Authors:

Title: Are reactive oxygen species (ROS) the main mechanism by which copper ion treatment degrades the DNA of Mycobacterium avium subsp. Paratuberculosis suspended in milk?

Comments

The submitted manuscript discussed the effect of copper ions, and ROS generated in response to oxidative stress, on the damage to Mycobacterium avium subsp. Paratuberculosis (MAP) DNA when exposed to a copper ion challenge in cow's milk. The spiked milk with different MAP bacterial loads was supplemented with blocking agents. These were either the copper chelators EDTA and BCS or the reactive oxygen species (ROS) quenchers D-mannitol, gallic acid and quercetin. DNA protection, MAP viability and ROS production generated after exposure to a copper challenge were then measured. In a bacterial load of 104 cells mL-1, both copper chelators and all ROS quenchers offered significant protection to MAP DNA. In a concentration of 102 cells mL-1, only D-mannitol and a mix of quenchers significantly protected the viability of the bacteria, and only at a concentration of 106 cells mL-1 was there a lower production of ROS when supplementing milk with gallic acid, quercetin, and mix of quenchers.

Despite the presented results, I think the submitted work is preliminary and does not merit publication. The authors should discuss the applicability of using copper on an industrial scale to fight the mycobacterium. Also, they should investigate in depth the mechanism of copper targeting the mycobacterium through different assays related to reactive oxygen species production.